# S-ABA Enhances Rice Salt Tolerance by Regulating Na^+^/K^+^ Balance and Hormone Homeostasis

**DOI:** 10.3390/metabo14040181

**Published:** 2024-03-23

**Authors:** Wenxin Jiang, Xi Wang, Yaxin Wang, Youwei Du, Shuyu Zhang, Hang Zhou, Naijie Feng, Dianfeng Zheng, Guohui Ma, Liming Zhao

**Affiliations:** 1College of Coastal Agriculture Sciences, Guangdong Ocean University, Zhanjiang 524088, China; 2112104057@stu.gdou.edu.cn (W.J.); 2112204057@stu.gdou.edu.cn (X.W.); 2112104017@stu.gdou.edu.cn (Y.W.); 2112104012@stu.gdou.edu.cn (Y.D.); zhangshuyu@stu.gdou.edu.cn (S.Z.); 21110901000051@hainanu.edu.cn (H.Z.); zhengdf@gdou.edu.cn (D.Z.); 2South China Center of National Saline-Tolerant Rice Technology Innovation, Zhanjiang 524088, China; 3State Key Laboratory of Hybrid Rice, Hunan Hybrid Rice Research Center, Changsha 410125, China

**Keywords:** rice, salt stress, S-ABA, Na^+^/K^+^ stability, hormone homeostasis

## Abstract

In order to explore the regulating role and the physiological and biochemical mechanisms of trans-abscisic acid (hereinafter referred as S-ABA) in the process of rice growth and development under salt stress, we took Chaoyou 1000 and Yuxiangyouzhan as materials and set up three salt concentration treatments, CK0 (Control treatment), N1 (50 mmol L^−1^ NaCl), and N2 (100 mmol L^−1^ NaCl), in potted trials; we aimed to study the mechanism of rice’s response to salt stress from the perspective of agricultural traits and physiological biochemicals and to improve rice’s resistance to salt stress through exogenously applying the regulating technology of S-ABA. The following results were obtained: Under salt stress, the growth of rice was significantly suppressed compared to CK0, exhibiting notable increases in agricultural indicators, photosynthesis efficiency, and the NA^+^ content of leaves. However, we noted a significant decrease in the K^+^ content in the leaves, alongside a prominent increase in NA^+^/K^+^ and a big increase in MDA (malondialdehyde), H_2_O_2_ (hydrogen peroxide), and O_2_^−^ (superoxide anion). This caused the cytomembrane permeability to deteriorate. By applying S-ABA under salt stress (in comparison with salt treatment), we promoted improvements in agronomic traits, enhanced photosynthesis, reduced the accumulation of NA^+^ in leaves, increased the K^+^ content and the activity of antioxidant enzymes, and reduced the active oxygen content, resulting in a sharp decrease in the impact of salt stress on rice’s development. The application of S-ABA decreased the endogenous ABA (abscisic acid) content under salt stress treatment but increased the endogenous GA (gibberellin) and IAA (indole acetic acid) contents and maintained the hormonal homeostasis in rice plants. To summarize, salt stress causes damage to rice growth, and the exogenous application of S-ABA can activate the pouring system mechanism of rice, suppress the outbreak of active oxygen, and regulate NA^+^/K^+^ balance and hormone homeostasis in the blades, thus relieving the salt stress.

## 1. Introduction

In recent years, under the influence of human activities and climate change, rising sea levels have intensified seawater’s intrusion into groundwater systems in coastal areas. Soil salinization in coastal areas has become increasingly serious, reducing the nutrient cycling and productivity of the land [1]. At present, about 20% of global arable land resources are made up of salinized soil [2]. Salinized soil in China has the characteristics of large area and high salination [3]. Saline–alkaline land is a scarce and reserved land resource with immense potential for comprehensive exploitation; therefore, the utilization of saline–alkaline land and an increase in rice production from salinized land are crucial for ensuring food security in China.

Rice (*Oryza sativa* L.) is more sensitive to salt than other grain crops, and salt stress mainly causes osmotic stress and ionic toxicity, which leads to various stress responses, such as morphological and physiological changes [4,5]. Osmotic stress mainly occurs in the early stage of salt stress and inhibits rice’s ability to absorb water and other nutrients, resulting in adverse effects on rice growth. Ionic toxicity, which is mostly due to accumulation of excessive Na^+^ and Cl^−^, further destroys the osmotic pressure and water potential in the cell membrane and damages the integrity of the cell membrane. As a result, the growth and development of rice is affected [6]. The entry of a high concentration of salt into rice plants will eventually increase the toxicity level of adult leaves and cause leaf senescence, which is the most direct consequence of cell death in plants and is characterized by synergistic changes at the physiological and molecular level (such as chlorophyll degradation, photosynthesis obstruction, membrane integrity damage, ROS and MDA accumulation, etc.) [7]. Excessive accumulation of ROS and the destruction of Na^+^/K^+^ stability are also important reasons for presenility of leaves [8]. Salt stress increases ROS levels, such as H_2_O_2_ and O_2_^−^, which are considered signaling substances regulating the aging process in leaves [9,10]. Excessive Na^+^ in the cytoplasm severely interferes with the absorption and transportation of K^+^, which plays an important role in enhancing salt tolerance and delaying leaf senescence [11].

Abscisic acid, also known as natural abscisic acid, is one of the molecular signaling plant hormones that can regulate plant growth, improve the quality of agricultural products, and enhance the survival ability of crops under adverse stress. In recent years, studies have shown that exogenous application of S-ABA can reduce water consumption, wilt degree, and transpiration rate by mediating stomatal closure [12]; On the other hand, it can help reduce Na^+^/K^+^ in plants, induce the accumulation of osmotic substances such as soluble sugar, alleviate the stress caused by osmotic pressure differences and various ion stresses, and maintain the relative integrity of cell membrane, resulting in a reduction salt damage [13]. Exogenous ABA can alleviate the damage to the photosynthetic system caused by stress, maintain the integrity of photosynthetic organs, increase the photosynthetic rate, and enhance the salt tolerance of plants [14].

Studies on the alleviation of exogenous ABA on rice under salt stress have thus far focused on the seedling stage of rice. For example, it has been reported that exogenous ABA application under salt stress significantly reduced root damage to rice seedlings, increased biomass accumulation and the survival rate of rice seedlings, increased the relative water content of rice seedlings, and reduced cell membrane damage and Na^+^/K^+^ ratio [15]. The study of Liu et al. [16] showed that ABA treatment significantly reduced the degradation of chlorophyll, inhibited the transcription of aging-related genes, delayed the aging of leaves, and enhanced tolerance to salt stress by inhibiting the accumulation of H_2_O_2_ in cells. In field experiments, the application of ABA significantly reduced leaf wilt and seedling death and improved the growth of rice plants under salt and alkali stress [17].

Most studies have only discussed the ability to improve rice seed germination and seedling salt tolerance, but the physiological mechanism of salt tolerance in the later stages of rice growth has been studied less frequently. On this basis, this study—through exploring the effects of exogenous S-ABA on rice growth and development, photosynthesis, activation of the antioxidant defense system, ion homeostasis, absorption and transport, and endogenous hormone homeostasis under salt stress—revealed the mechanism involved in exogenous S-ABA promoting rice’s resistance to salt stress, aiming to provide a theoretical basis for further research on the improvement of rice yield in saline–alkaline areas using exogenous S-ABA.

## 2. Materials and Methods

### 2.1. Design and Program of the Experiment

(1) Experimental materials: The tested rice varieties were Chaoyou Qianhao (CYQH) and Yuxiang Youzhan (YXYZ), which were provided by Hunan Hybrid Rice Research Center and Guangdong Rice Research Institute, respectively. The test agent was S-ABA, with the effective ingredient at a volume of 0.03% (S-ABA, trade name “Yakexi”) provided by Jiangxi Xinruifeng Biochemical Co., Ltd. (Jian, China). The experiment was conducted in the daylight multi-greenhouse of Guangdong Ocean University from May to August in 2023.

(2) Seedling cultivation: under no salt stress, conventional methods were used to soak the seeds, promote germination, and raise the seedlings. The seedlings were planted manually when they reached the four-leaf stage, and the seedlings uniform growth were selected and transplanted into a watertight plastic basin (with an upper diameter of 27.5 cm, a lower diameter of 18.5 cm, and a height of 23 cm), and the water was regularly supplemented to maintain the water layer. The plastic basin was filled with 8 kg of soil, 4 hills per basin, and 2 plants per hill; the spacing between each hill was 10 cm, and the planting depth was 1.5 cm. The test soil was laterite.

(3) Salt stress and S-ABA treatment: Salt treatment was applied on the 5th day after the seedlings resumed their growth after transplanting, and CK0, N1 (50 mmol L^−1^), and N2 (100 mmol L^−1^) were set to simulate salt stress conditions. We sprayed 0.03% S-ABA (diluted 100 times) on the leaf surface at the young panicle differentiation stage (35 d after transplanting), and kept the soil wet without dripping. We set up a total of 6 treatments: CK0 (Control treatment), N1 (50 mmol L^−1^ NaCl), N2 (100 mmol L^−1^ NaCl), SN0 (water + S-ABA spray at the stage of young panicle differentiation), SN1 (50 mmol L^−1^ NaCl + S-ABA spray at the stage of young panicle differentiation), and SN2 (100 mmol L^−1^ NaCl + S-ABA), which were sprayed at the differentiation stage of young panicles. The experiment was conducted using a completely random design. Samples of the two and three leaves of each treatment were collected on the 15th day after treatment at the young panicle differentiation stage and stored in the refrigerator at −80 °C for further testing.

(4) Fertilization management: Conversion of the amount of fertilization was based on field fertilization method (amount per hectare). Base fertilizer was applied 2 days before transplantation, and 0.499 g of nitrogen fertilizer (urea), 0.7128 g of phosphorus fertilizer (superphosphate), and 0.4455 g of potassium fertilizer (potassium chloride) were used in each plastic basin, respectively. Tillering fertilizer was applied one week later, and 0.552 g of nitrogen fertilizer was applied per basin. Fertilizer was added regularly according to the seedling condition, and pesticides was sprayed regularly to prevent insects and diseases. Among them, the ratio of the amount of nitrogen fertilizer was as follows: base fertilizer–tillering fertilizer–panicle fertilizer = 4:3:3. Phosphate fertilizer was used as the base fertilizer, and the ratio of the amount of potassium fertilizer was base fertilizer–panicle fertilizer = 1:1. At the tillering stage, 0.70 g urea and 0.56 g potassium chloride were applied at the panicle differentiation stage.

### 2.2. Measurement of Growth Characteristics

#### Measurement of Morphological Indicators

On the 15th day after S-ABA treatment, 8 rice plants with uniform growth were randomly selected to measure seedling height, stem diameter and leaf area (2 and 3 leaves of the main stem were reversed), then stored at 105 °C for 30 min and dried at 80 °C until reaching a constant weight. We used their dry matter weight thereafter.

### 2.3. Gas Exchange Parameters and Measurement of SPAD

The net photosynthetic rate (Pn), stomatal conductance (Gs), intercellular CO_2_ concentration (Ci), transpiration rate (Tr), and other photosynthetic parameters were measured from 09:00 a.m. to 11:30 a.m using a LI-6800XT portable photosynthetic measurement system produced by the LI-COR company of the United States. The measurements were repeated 8 times for each treatment. The photosynthetically active radiation was set at 1000 μmol·m^−2^·s^−1^ and the CO_2_ concentration was 400 μmol·mol^−1^.

The chlorophyll content of leaves was measured using a SPAD 502 Plus chlorophyll analyzer; each sample was analyzed in triplicate.

### 2.4. ROS Content Determination

The hydrogen peroxide (peroxide, H_2_O_2_) content was determined by referring to the potassium iodide method described by Feng et al. [18].

Superoxide radical O_2_^−^ was detected using a plant superoxide anion (SOA) ELISA kit. The kit uses a two-antibody one-step sandwich enzyme-linked immunosorbent assay (ELISA). Absorbance (OD value) was measured by enzyme-labeled instrument at a 450 nm wavelength, and the sample concentration was calculated.

The content of malondialdehyde (MDA) was determined by referring to Di et al. [19]. Absorbance was recorded at 450 nm, 532 nm, and 600 nm.

### 2.5. Measurement of Antioxidant Enzyme Activity

For measurement of superoxide dismutase (SOD) and peroxidase (POD; EC 1.11.1.7), we referred to Jiang et al. [20]; for catalase (CAT; EC 1.11.1.6), we referred to Panda et al. [21]. For measurement of the activity of ascorbate peroxidase (APX), we referred to Gratao et al. [22].

### 2.6. Measurement of Non-Antioxidant Content

The content of soluble sugar (SS) was measured according to the method described by Feng et al. [18]. Soluble protein (SP) content was measured using the Thomas Brilliant Blue G-250 method described by Jiang et al. [23]. The reduced ascorbate (ASA) content was measured according to the method given by Takumi et al. [24]. The method described by Tasir S et al. was used to quantify GSH concentrations [25].

### 2.7. Determination of Ascorbic Acid (AsA) and Glutathione (GsH) Levels

Ascorbic acid (AsA) levels were determined using methods described by Altaf [26] with minor modifications. The absorbance was measured at 534 nm. The GsH content was determined as described by Khairiah Mubarak Alwutayd [27] with minor modifications. The reaction mixture consisted of 200 µL of supernatant, 2.6 mL of acetate buffer (0.2 M, pH 5.6), and 200 µL of 5,5-disulfuric acid-(2-nitrobenzoic acid). Samples were stored at 30 °C for 10 min. The absorbance of GsH was measured at 412 nm.

### 2.8. Endogenous Hormone Level of the Plants

Abscisic acid (ABA), gibberellin (GA), and indole acetic acid (IAA) were measured by colorimetry using an ELISA plant detection kit according to the instructions provided (Shanghai Nibe Biotechnology Co., Ltd., Shanghai, China). The kit uses a two-antibody one-step sandwich enzyme-linked immunosorbent assay (ELISA). Specimens, standards, and HRP-labeled detection antibodies were added to microwells pre-coated with hormone abscisic acid (ABA), gibberellin (GA), and indole acetic acid (IAA) antibodies; they were then incubated and washed thoroughly. We used the chromogenic substrate TMB, which is catalyzed by peroxidase and converted to its final yellow color by acid. There was a positive correlation between the color and the hormone abscisic acid (ABA), gibberellin (GA), and indole acetic acid (IAA) in the sample.

### 2.9. Measurement of Na^+^ and K^+^ Content

The rice leaves of each treatment were weighed dry and cut into small pieces of 5–10 mm with scissors. After cleaning up as the first step, the dry samples of the leaves were dissolved with a mixture of HNO_3_:HClO_4_ (2:1, *v*/*v*). After that, the volume of deionized water was fixed to 100 mL. The absorbance values of the leaf samples were measured using a FP6410 flame photometer, and the contents of Na^+^ and K^+^ in the samples were calculated.

### 2.10. Statistical Analysis

Excel 2019 was used for original data processing. Statistic software SPSS 25.0 was used to conduct a one-way ANOVA analysis and multi-dimensional Duncan comparison. Additionally, Microsoft Excel 2019 and Origin 9.1 software was used for graphing.

## 3. Results and Analysis

### 3.1. Effects of S-ABA Treatment on Growth and Development of Rice under Salt Stress

As can be seen from Table 1, salt stress significantly inhibited the growth characteristics of the two genotypes of rice varieties. S-ABA treatment can significantly improve the growth characteristics of rice under salt stress and reduce stress toxicity. Salt treatment significantly (*p* < 0.05) decreased the plant height, stem diameter, leaf area and dry weight of two rice varieties, and compared with CK0 treatment, the N1 treatment decreased the above factors of Chaoyou1 qianhao by 9.18%, 16.32%, 8.30%, and 17.18%. The values of Yuxiang youzhan decreased by 17.82%, 21.28%, 37.55%, and 41.05%,respectively. For N2 treatment, decreases of 17.78%, 24.39%, 37.55%, and 41.05%, respectively, and 16.50%, 39.25%, 51.75%, and 61.40%, respectively, were found. For SN1 treatment, increases of 9.52%, 18.39%, 19.15%, and 18.28%, for plant height, stem diameter, leaf area, and dry matter weight were found (and 14.29%, 29.79%, 35.80%, and 79.30%, respectively). For the SN2 treatment, increases of 13.13%, 24.04%, 27.69%, 34.32% and 11.78%, 36.38%, 53.37%, and 114.21%, respectively, were found. The results indicated that S-ABA treatment could promote the accumulation of above-ground material, promote plant growth, and enhance resistance to salt stress.

### 3.2. Effects of Exogenous S-ABA Treatment on Na^+^/K^+^ Homeostasis and Osmotic Substance Regulation under Salt Stress

Maintaining a low level of Na^+^/K^+^ plays an important role in plant salt tolerance, so the contents of Na^+^ and K^+^ in rice leaves of two genotypes were studied. Under salt stress, the accumulation of Na^+^ content in the two rice varieties increased sharply, while the accumulation of K^+^ decreased significantly, increasing the ratio of Na^+^/K^+^ (Figure 1). Exogenous application of S-ABA could reduce the content of NA^+^ at different degree, increase the content of K^+^, and decrease the ratio of Na^+^/K^+^. Under N1 treatment, the accumulation of Na^+^ in CYQH and YXYZ increased by 16.67% and 164.44, respectively, and the accumulation of K^+^ decreased by 17.21% and 16.05%, respectively. Under N2 treatment, Na^+^ increased by 125.93% and 457.78%, and K^+^ decreased by 18.32% and 27.40%, respectively. After S-ABA treatment, the contents of CYQH and YXYZ Na^+^ decreased by 12.70% and 9.17% (SN1), 19.67% and 62.15% (SN2), and K^+^ increased by 6.74%, 12.52% and 8.06% and 55.32% (SN1), respectively. Under salt stress, Na^+^/K^+^ of both genotypes was significantly increased, while exogenous S-ABA treatment significantly decreased the Na^+^/K^+^ content of both genotypes and SN1 treatment decreased the Na^+^/K^+^ content by 25% (CYQH) and 33.33% (YXYZ), respectively. The SN2 treatment decreased it by 28.57% (CYQH) and 73.68% (YXYZ), respectively. The results showed that exogenous application of S-ABA could reduce ion stress induced by salt stress in rice.

Osmotic regulation is an important mechanism through which plants cope with stress; it can regulate the leaf transpiration rate and maintain physiological effects including normal photosynthesis [28]. Under salt stress, plant cells stabilize the cell membrane structure through the production of soluble protein, soluble sugar, and other small molecular organic matter so as to reduce the damage salt stress causes to plants. As shown in Figure 2, compared with CK0, the SP and SS contents in CYQH showed an increasing trend under salt stress, while the SP and SS contents in YXYZ showed a decreasing trend. The SP and SS contents of the two rice varieties were significantly increased after the application of S-ABA under salt stress, indicating that S-ABA can alleviate salt damage by increasing the content of osmotic substances in rice leaves under salt stress and enhancing the stability of cell membrane; it can also maintain normal osmotic pressure and water potential in plant cells under stress, thus enhancing the salt tolerance of rice.

### 3.3. Effects of Exogenous S-ABA Treatment on Photosynthetic Efficiency and SPAD Value of Rice under Salt Stress

As shown in Table 2, osmotic stress induced by N2 salt treatment resulted in a significant decrease in Pn, Gs, Tr and Ci of the two rice varieties compared with CK0, and the respective percentage decreases are 37.07%, 70.00%, 68.80%, 15.10% (CYQH), and 32.42%, 62.96%, 45.17%, 14.38% (YXYZ). The gas exchange parameters were improved significantly through the application of S-ABA under salt stress. SN1 treatment increased Pn, Gs and Tr by 24.13%, 76.47%, and 28.18% for CYQH and 22.69%, 29.17%, 27.78% for YXYZ, respectively. Additionally, SN2 treatment increased them by 38.00%, 90.48%, and 105.61% CYQH and by 32.42%, 43.33%, 29.57% in YXYZ, respectively. Under salt stress, the SPAD values of the two rice varieties were shown to significantly decrease; of these, CYQH’s SPAD value decreased significantly less than that of the YXYZ variety, indicating that when rice plants were subjected to salt stress, chlorophyll synthesis was blocked, decomposition was accelerated, and the photosynthetic pigment content in leaves was reduced. Exogenous S-ABA treatment significantly increased the SPAD values of the two rice varieties, indicating that S-ABA was conducive to maintaining high photosynthesis in and material production of rice.

### 3.4. Effects of Exogenous S-ABA Treatment on Alleviating Physiological Mechanisms under Salt Stress

As shown in Figure 3, salt stress and application of S-ABA significantly affected the activity of antioxidant enzymes in the two rice varieties. Compared with CK0, the SOD activity of CYQH was significantly decreased under salt stress, while the SOD activity of plants under S-ABA was significantly increased. The SOD activity of YXYZ was significantly increased under salt stress, and S-ABA treatment further increased SOD activity. Similarly, the POD and APX activities of the two rice varieties treated with S-ABA under salt stress were significantly higher than those treated without S-ABA. Compared with salt treatment, S-ABA treatment under salt stress significantly decreased the CAT activity of rice (CYQH) and increased the CAT activity of rice (YXYZ). Compared with SN0, the SN1 treatment showed no significant difference, but the SN2 treatment induced significant increases in POD, CAT and APX activities, by 40.45%, 90.08%, and 60.80% respectively. These results indicate that the rice plants that underwent S-ABA treatment showed higher salt tolerance, thus maintaining their cell physiology and enzyme stability under salt stress.

As shown in Figure 4,compared to CK0, significant increases in levels of H_2_O_2_ (from 11.74% to 40.97% for CYQH and from 23.61% to 26.30% for YXYZ) and O_2_^−^ (from 11.6% to 20.68% for CYQH and from 13.28% to 21.42% for YXYZ) in the leaves of the two rice varieties were shown under salt stress. The effects of S-ABA caused a prominent improvement in H_2_O_2_ and O_2_^−^ contents under salt stress, decreasing by 13.23–17.95% and 41.16–44.90% (CYQH) and 24.38–27.23% and 20.22–25.15% (YXYZ), respectively. MDA contents reflected the integrity of leaf cell membranes and the peroxidation levels of membrane plasma. Under salt stress, MDA content was significantly increased, and compared with CK0, it was increased by 19.71–77.37% for CYQH and 54.75–79.33% for YXYZ. MDA content decreased significantly after S-ABA application; it decreased by 18.29% to 24.28% for CYQH and by 21.66% to 28.66% for YXYZ compared with salt treatment without S-ABA application. These results indicate that S-ABA has strong antioxidant ability and has a significant effect on ROS clearance.

As shown in Figure 5, compared with CK0 treatment, salt stress significantly increased the AsA content of CYQH, but had no significant effect on YXYZ. Under salt stress, the spraying of S-ABA significantly increased the AsA content of two rice varieties. A bigger increase was observed in the SN2 treatment (by 83.46% and 258.38%, respectively). Under salt stress, the GsH content of CYQH was significantly decreased and the GsH content of YXYZ was significantly increased. Under salt stress, spraying of S-ABA significantly decreased the GsH content of CYQH and significantly increased the GsH content of YXYZ. The results showed that the AsA content of both rice varieties showed an increasing trend under salt stress. The AsA content of rice plants was further increased and the salt tolerance of rice was enhanced by spraying S-ABA, and the decrease in the antioxidant content of rice was inhibited by spraying S-ABA, indicating that spraying plants with S-ABA can effectively help them to cope with oxidative damage under salt stress.

### 3.5. Regulation of Endogenous Hormone Stability in Rice Leaves from Applying Exogenous S-ABA under Salt Stress

Salt stress and leaf spraying of S-ABA significantly changed endogenous ABA, GA, and IAA levels in the two rice varieties(Figure 6). Compared with CK0, salt stress significantly reduced the ABA content of the two rice varieties, and S-ABA spraying under salt stress had no significant effect on the ABA content of CYQH. Under salt stress, the ABA content of rice treated with S-ABA decreased more significantly than that of rice treated without S-ABA. Compared with CK0 treatment, salt stress significantly increased the GA and IAA contents of the two rice varieties, and spraying with S-ABA under the conditions of salt stress further increased the GA and IAA contents of the two rice varieties. These results suggest that S-ABA plays an important role in balancing endogenous hormone levels in rice plants under salt stress.

### 3.6. Correlation Analysis of Physiological and Biochemical Indexes

To more effectively identify the correlation between these traits, a Pearson correlation analysis was performed for 25 representative traits (Figure 7). For the Chaoyou 1000 variety, there was a significant positive correlation between agronomic traits and photosynthetic parameters. There was also a positive correlation between agronomic traits, photosynthetic parameters, and endogenous hormone ABA content, and a significant positive correlation between photosynthetic parameters and K^+^ content. SP, SS, POD, APX, ASA were significantly positively correlated with GA and IAA. Agronomic traits and photosynthetic parameters were significantly negatively correlated with MDA, O_2_^−^, CAT activity, NA^+^ and Na^+^/K^+^.

For Yuxiangyouzhan, there was a significant positive correlation between the agronomic traits and photosynthetic parameters. SP and SS were positively correlated with ASA. Agronomic traits, photosynthetic parameters and SPAD values were significantly and negatively correlated with MDA, NA^+^ and Na^+^/K^+^, and agronomic traits were significantly negatively correlated with O_2_^−^ too.

## 4. Discussion

As salt stress is one of the major abiotic stresses that adversely affect rice growth, in-depth studies on plants’ phenotypes, response mechanisms, and repair strategies have been carried out [29]. Salt stress severely restricts the growth and development of rice, and different varieties of the same plant have different responses to salt stress [30,31,32,33,34]. As an important metabolic and synthetic organ of plants, the growth condition of the above-ground part directly reflects the salt tolerance of plants, and the better the growth of the above-ground part, the higher the salt tolerance capacity of plants. In addition, as the main location of photosynthesis in plants, the leaf area not only reflects the growth of plants but also reflects their photosynthetic efficiency [35]. Declines in growth-related attributes due to salt stress have previously been reported in a variety of crops. The reason for the significant inhibition of growth parameters under salt stress is that the rapid accumulation of Na^+^ in plants breaks the ion balance and metabolic level of the normal growth of plants, eventually leading to serious inhibition of growth [31]. Therefore, more and more studies have focused on improving the salt tolerance of rice and its related mechanisms; however, its role in regulating rice panicle differentiation remains unclear. The results of this study support our hypothesis that exogenous S-ABA can promote the salt tolerance of rice at different salt concentrations. The responses of rice plant growth to salt stress were assessed with respect to photosynthetic characteristics, ROS levels, antioxidant activity, endogenous hormone homeostasis, and Na^+^/K^+^ homeostasis.

### 4.1. Exogenous S-ABA Maintains Normal Rice Growth under Salt Stress

Under salt stress, Na+ is absorbed by rice roots, which inhibits water retention ability and delays root growth, thus affecting the growth of the above-ground parts of rice [36]. The results showed that salt stress significantly inhibited the plant height, stem diameter, leaf area and dry matter weight of the two rice varieties, and S-ABA application under salt stress could improve plant growth and biomass and enhance the salt tolerance of rice.

### 4.2. Exogenous S-ABA’s Effects on Na^+^/K^+^ Stability in Rice under Salt Stress

Several results have shown that the damage to Na^+^/K^+^ stability caused by salt stress is the main factor inhibiting plant growth [37]. NA^+^ toxicity affects the ability to maintain the acquisition and distribution of K^+^ in plants [38]. The results of this experiment confirmed that Na^+^/K^+^ homeostasis is a key factor in plants’ salt tolerance. The application of S-ABA maintained Na^+^/K^+^ homeostasis in both rice varieties under salt stress. K^+^ plays an important role in osmotic regulation and enzyme activation, thereby preventing leaf aging and improving salt tolerance [39]. In this study, K^+^ was decreased significantly under salt stress, and this decrease was significantly alleviated after the application of S-ABA. It is worth noting that in this study, different responses of the two rice varieties to salt stress were shown, and responses to different salt concentrations were also inconsistent. The degree of decrease in K^+^ in the CYQH variety was significantly higher than that in the YXYZ variety, and the degree of increase in K^+^ in the CYQH variety was significantly higher than that in the YXYZ variety, even after S-ABA application. This suggests that S-ABA has different mechanisms through which it regulates salt tolerance under different salt concentrations and varieties, and S-ABA may give priority to antioxidant protection rather than ion homeostasis in order to enhance salt tolerance under lower salt concentrations.

### 4.3. Exogenous S-ABA’s Effects on Photosynthetic Efficiency in Rice under Salt Stress

Existing studies have shown that rice plants are prone to premature aging under salt stress. In the process of leaf aging, the most significant phenotypic characteristic is the yellowing of functional leaves caused by the slowing down of chlorophyll structure cracking and the chlorophyll synthesis rate, which are accompanied by the weakening of photosynthesis [40]. Chlorophyll is the most important factor in the photosynthesis of green plants. A high content of chlorophyll is conducive to the utilization of light energy in plants, but chlorophyll is very vulnerable to ROS damage. In this study, it was found that SPAD content decreased significantly under salt stress, inferring that the growth of glume shells during panicle differentiation was related to the decrease in chlorophyll content in leaves; this may be because more synthesized chlorophyll was distributed to newborn panicle shells. Pn, Gs, and Tr all decreased significantly, and the main reason for the decrease in Pn may be stomatal limitation. The lower root water potential under salt stress and the decrease in ABA content transported from root to the ground would induce stomatal closure [41]. A similar result was obtained in this experiment. As an anti-transpiration agent, S-ABA can induce stomatal closure and increase chlorophyll content so as to improve the photosynthetic ability of rice [42]. In this study, it was proved that S-ABA inhibited photosynthetic degradation of two rice varieties under salt stress, significantly increasing Pn value and maintaining a high level of SPAD in two rice varieties under salt stress. The results showed that S-ABA effectively slowed down premature aging and photomorphogenesis of rice leaves under salt stress.

### 4.4. Exogenous S-ABA’s Effects on Photosynthetic Efficiency in Rice under Salt Stress

Existing research results suggest that the damage caused by salt stress to rice is mainly because of the sharp accumulation of ROS and MDA [43]. Excessive accumulation of ROS will trigger oxidative stress response in plants, resulting in damage to cell membranes and even cell death, accompanied by chlorophyll degradation and low photosynthesis efficiency. The ROS induced by salt stress was mainly composed of H_2_O_2_ and O_2_^−^ [44]. The results showed that a significant amount of ROS was accumulated in both rice varieties under salt stress. As the most important endogenous hormone in plants, ABA is involved in the antioxidant defense response of plants and regulates the response of plants to stress [15]. In this study, the H_2_O_2_ production rate and O_2_^−^ content in the leaves of two rice varieties treated with S-ABA under salt stress were significantly lower than those without S-ABA treatment, which further indicated the scavenging effect of S-ABA on ROS.

The activities of SOD, POD, CAT and APX in the leaves of different rice varieties exhibited changes under different salt concentrations (Figure 3). The results showed that under salt treatment, the SOD activity of CYQH rice leaves decreased significantly compared with CK0, while the activities of POD, CAT and APX in CYQH rice and SOD, POD, CAT, and APX in YXYZ all increased sharply compared with CK0. The enzyme activity of CYQH rice leaves increased significantly compared with CK0 under high salt concentrations. This further indicated that high concentrations of salt cause great damage to the active oxygen-scavenging system. However, S-ABA promotes ROS clearance by further enhancing antioxidant activity in rice leaves. Conversely, the maintenance of membrane integrity and a reduction in lipid peroxidation in cell membranes were caused by these oxidative stress responses under salt stress. The decrease in MDA, H_2_O_2_ and O_2_^−^ contents in leaves also proved this. Therefore, the results of this study indicate that S-ABA can alleviate damage caused to rice under salt stress by blocking the outbreak of active oxygen and enhancing the activity of antioxidant enzymes, thus enhancing the salt tolerance of rice.

Non-enzymatic antioxidant defense mechanisms involve different osmotic substances (amino acids, sugars, etc.) [45]. AsA and GsH, as important non-enzymatic antioxidants, participate in many different metabolic processes of plants and directly affect plant growth [46,47]. In this study, AsA and soluble sugar content were shown to accumulate in rice treated with S-ABA under salt stress. Previously, exogenous application of ABA under salt stress significantly increased non-enzymatic defense mechanisms (ascorbic acid (AsA), glutathione (GsH), carotenoids, and flavonoids) in response to oxidative stress [48]. In summary, the results of this study clearly indicate that exogenous S-ABA can effectively alleviate the oxidative damage to rice caused by salt stress by activating multiple defense mechanisms of rice.

### 4.5. Exogenous S-ABA’s Effects on Endogenous Hormone Dynamics in Rice under Salt Stress

Plant hormones, including abscisic acid, gibberellin and auxin, are involved in the stress response of plants under stress conditions, accelerating growth and development through local or overall action [49]. This study showed that the application of S-ABA could significantly increase GR and IAA levels to alleviate salt stress (Figure 6). However, leaf spraying of S-ABA significantly reduced the ABA level of rice plants. This may be due to the direct interaction of small-molecule signaling components of various hormones in various physiological processes. Different hormone signal transduction pathways coordinate with each other to balance the complex regulatory mechanisms and ultimately improve the response of plants to different stress challenges.

The correlation analysis of rice representative indicators showed that plant height, leaf area, and dry matter weight were significantly or extremely significantly positively correlated with photosynthetic parameters, SPAD, ABA, and K^+^ contents; however, they were significantly or extremely significantly negatively correlated with MDA, H_2_O_2_, O_2_^−^, POD activity, CAT activity, and Na^+^/K^+^ contents in leaves. The results indicated that the application of S-ABA under salt stress could improve the photosynthetic capacity and antioxidant enzyme activity of rice; inhibit the increase in MDA, H_2_O_2_ and O_2_^−^ contents in leaves; maintain the stability of Na^+^/K^+^ in leaves; and alleviate the osmotic stress in rice under salt stress so as to improve the salt tolerance of rice.

## 5. Conclusions

Salt stress can inhibit the growth of rice, affect the photosynthetic capacity of leaves, break ion homeostasis in plants, and destroy the homeostasis balance of Na^+^/K^+^ and endogenous hormones. S-ABA enhances ion homeostasis and the high K^+^ content of rice under salt stress, which is conducive to the enhancement of antioxidant enzyme activity. The enhancement of antioxidant enzyme activity and the antioxidant effect of S-ABA itself ensure the clearance of ROS, reducing the damage caused by oxidative stress and the degradation of chlorophyll (i.e., the SPAD value). In addition, the high content of chlorophyll is conducive to the utilization of light energy and full photosynthesis in leaves. Exogenous S-ABA promoted the salt tolerance of rice by improving the photosynthetic capacity and physiological activity of functional leaves of rice. However, it varied greatly depending on salt concentration and rice variety. When under a relatively low salt concentration (50 mmol L^−1^ NaCl), S-ABA regulated the salt tolerance of rice mainly by improving the antioxidant system; under a higher salt concentration (100 mmol L^−1^ NaCl), S-ABA regulated the salt tolerance of rice mainly by regulating the homeostatic balance of Na^+^/K^+^ and the stability of hormones, thereby reducing the damage caused by salt stress to rice.

## Figures and Tables

**Figure 1 metabolites-14-00181-f001:**
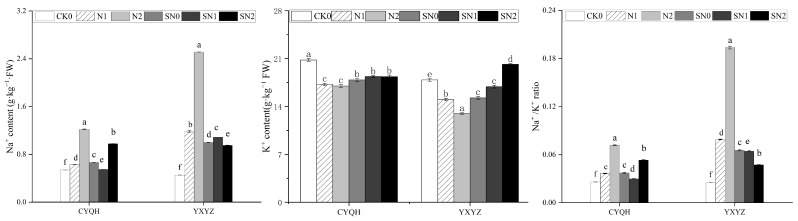
Effect of S-ABA on ionic homeostasis in rice under salt stress. CK0 (Control treatment), N1 (50 mmol L^−1^ NaCl), N2 (100 mmol L^−1^ NaCl), SN0 (water + S-ABA spray at the stage of young panicle differentiation), SN1 (50 mmol L^−1^ NaCl + S-ABA spray at the stage of young panicle differentiation), SN2 (100 mmol L^−1^ NaCl + S-ABA). Different lowercase letters indicate significant differences between different treatments of the same rice variety (*p ≤* 0.05).

**Figure 2 metabolites-14-00181-f002:**
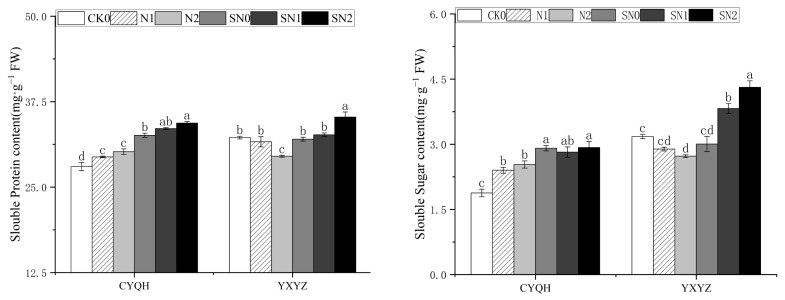
Effect of S-ABA on osmotic substances in rice under salt stress. CK0 (control treatment), N1 (50 mmol L^−1^ NaCl), N2 (100 mmol L^−1^ NaCl), SN0 (water + S-ABA spray at the stage of young panicle differentiation), SN1 (50 mmol L^−1^ NaCl + S-ABA spray at the stage of young panicle differentiation), SN2 (100 mmol L^−1^ NaCl + S-ABA). Different lowercase letters indicate significant differences between different treatments of the same rice variety (*p ≤* 0.05).

**Figure 3 metabolites-14-00181-f003:**
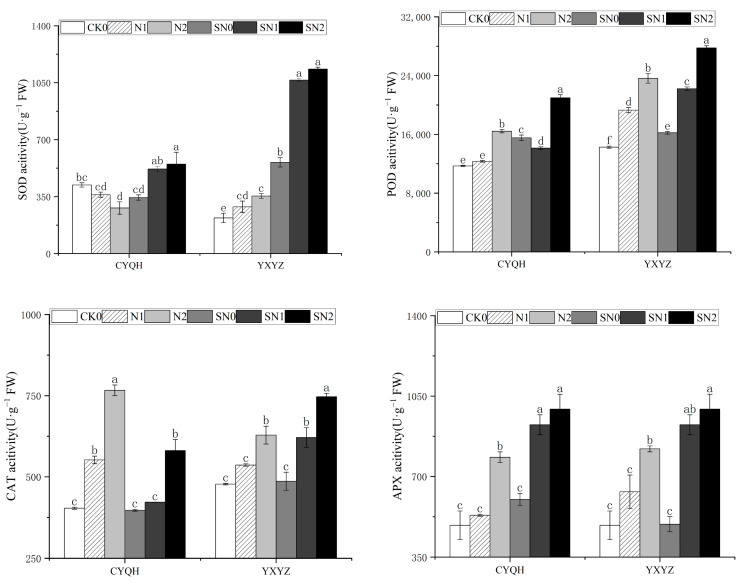
Effect of S-ABA on antioxidant enzyme activity in rice under salt stress. CK0 (control treatment), N1 (50 mmol L^−1^ NaCl), N2 (100 mmol L^−1^ NaCl), SN0 (water + S-ABA spray at the stage of young panicle differentiation), SN1 (50 mmol L^−1^ NaCl + S-ABA spray at the stage of young panicle differentiation), SN2 (100 mmol L^−1^ NaCl + S-ABA), Different lowercase letters indicate significant differences between different treatments of the same rice variety (*p* ≤ 0.05).

**Figure 4 metabolites-14-00181-f004:**
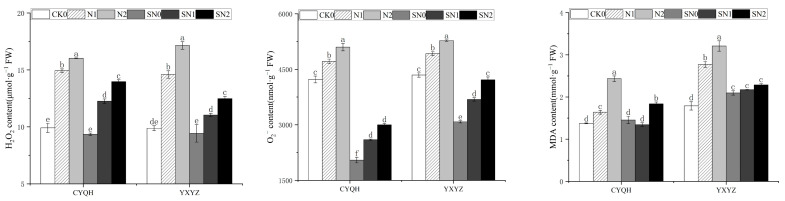
Effect of S-ABA on ROS content in rice under salt stress. CK0 (control treatment), N1 (50 mmol L^−1^ NaCl), N2 (100 mmol L^−1^ NaCl), SN0 (water + S-ABA spray at the stage of young panicle differentiation), SN1 (50 mmol L^−1^ NaCl + S-ABA spray at the stage of young panicle differentiation), SN2 (100 mmol L^−1^ NaCl + S-ABA). Different lowercase letters indicate significant differences between different treatments of the same rice variety (*p* ≤ 0.05).

**Figure 5 metabolites-14-00181-f005:**
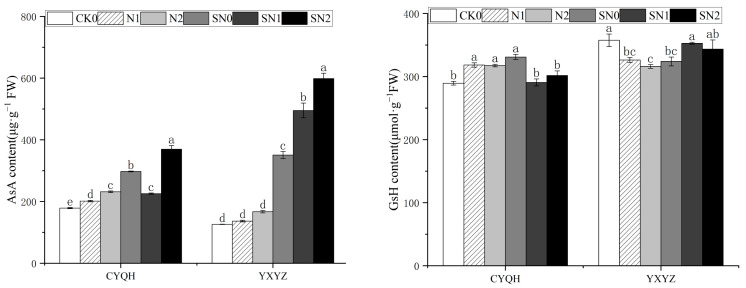
Effects of S-ABA on non-antioxidant content in rice under salt stress. CK0 (control treatment), N1 (50 mmol L^−1^ NaCl), N2 (100 mmol L^−1^ NaCl), SN0 (water + S-ABA spray at the stage of young panicle differentiation), SN1 (50 mmol L^−1^ NaCl + S-ABA spray at the stage of young panicle differentiation), SN2 (100 mmol L^−1^ NaCl + S-ABA). Different lowercase letters indicate significant differences between different treatments of the same rice variety (*p* ≤ 0.05).

**Figure 6 metabolites-14-00181-f006:**
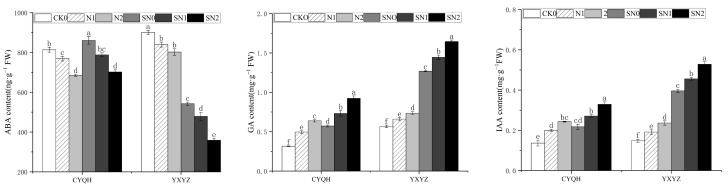
Effect of S-ABA on endogenous hormone content in rice under salt stress. CK0 (control treatment), N1 (50 mmol L^−1^ NaCl), N2 (100 mmol L^−1^ NaCl), SN0 (water + S-ABA spray at the stage of young panicle differentiation), SN1 (50 mmol L^−1^ NaCl + S-ABA spray at the stage of young panicle differentiation), SN2 (100 mmol L^−1^ NaCl + S-ABA). Different lowercase letters indicate significant differences between different treatments of the same rice variety (*p* ≤ 0.05).

**Figure 7 metabolites-14-00181-f007:**
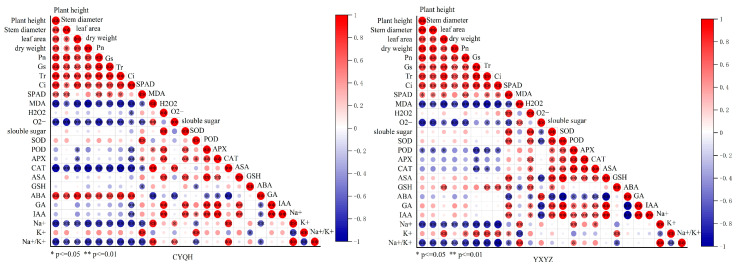
Correlation analysis of physiological traits in rice.

**Table 1 metabolites-14-00181-t001:** Effects of S-ABA on the morphological indices of different varieties of rice under salinity stress.

Treatments		Plant Height (cm)	Stem Diameter (mm)	Leaf Area (mm^2^)	Dry Matter Weight (g)
CYQH	CK0	74.4 ± 0.66 a	32.23 ± 0.52 b	3829.23 ± 41.81 bc	6.87 ± 0.39 a
	N1	67.57 ± 2.39 b	26.97 ± 0.92 c	3511.37 ± 145.12 c	5.69 ± 0.30 b
	N2	61.17 ± 0.67 c	25.37 ± 1.89 c	2391.27 ± 200.61 e	4.05 ± 0.37 c
	SN0	76.77 ± 0.82 a	36.70 ± 1.89 a	4357.67 ± 91.67 a	6.42 ± 0.12 ab
	SN1	74.00 ± 1.74 a	31.93 ± 0.33 b	4183.77 ± 121.39 ab	6.73 ± 0.41 a
	SN2	69.20 ± 1.25 a	31.47 ± 1.87 b	3053.37 ± 102.66 d	5.44 ± 0.16 b
YXYZ	CK0	75.93 ± 0.27 c	21.40 ± 0.96 bc	4645.97 ± 446.60 a	5.00 ± 0.63 bc
	N1	70.20 ± 0.51 d	18.80 ± 2.02 cd	2816.33 ± 187.93 cd	3.43 ± 0.48 cd
	N2	63.40 ± 0.16 e	13.00 ± 0.72 e	2250.90 ± 337.64 d	1.93 ± 0.11 d
	SN0	83.90 ± 0.21 a	27.80 ± 0.53 a	4258.63 ± 81.37 ab	8.25 ± 0.42 a
	SN1	80.23 ± 0.69 b	24.40 ± 0.17 b	3824.70 ± 239.30 ab	6.15 ± 0.25 b
	SN2	70.87 ± 0.80 d	17.73 ± 0.07 d	3452.23 ± 161.19 bc	4.07 ± 0.92 c

Note: CK0 (control treatment), N1 (50 mmol L^−1^ NaCl), N2 (100 mmol L^−1^ NaCl), SN0 (water + S-ABA spray at the stage of young panicle differentiation), SN1 (50 mmol L^−1^ NaCl + S-ABA spray at the stage of young panicle differentiation), SN2 (100 mmol L^−1^ NaCl + S-ABA). Different lowercase letters indicate significant differences between different treatments of the same rice variety (*p* ≤ 0.05).

**Table 2 metabolites-14-00181-t002:** Effect of S-ABA on photosynthetic parameters and SPAD values of different varieties rice under salt stress.

Treatments		Pn (μmol·m^−2^·s^−1^)	Gs (mmol·m^−2^·s^−1^)	Tr (μmol·m^−2^·s^−1^)	Ci (μmol·mol^−1^)	SPAD
CYQH	CK0	22.12 ± 1.26 ab	0.70 ± 0.05 a	10.29 ± 0.49 a	331.48 ± 2.74 a	42.07 ± 0.19 bc
	N1	16.41 ± 0.89 d	0.34 ± 0.04 bc	6.53 ± 0.38 c	322.34 ± 0.58 a	39.20 ± 0.25 d
	N2	13.92 ± 0.67 d	0.21 ± 0.03 c	3.21 ± 0.34 d	281.43 ± 5.34 d	36.83 ± 0.18 e
	SN0	23.90 ± 0.92 a	0.64 ± 0.06 a	10.31 ± 0.20 a	327.27 ± 1.70 a	41.67 ± 0.86 c
	SN1	20.37 ± 0.49 bc	0.60 ± 0.10 a	8.37 ± 0.83 b	313.55 ± 2.42 b	44.27 ± 1.58 ab
	SN2	19.21 ± 0.53 c	0.40 ± 0.05 b	6.60 ± 0.50 c	298.45 ± 1.90 c	45.17 ± 0.18 a
YXYZ	CK0	21.13 ± 0.60 ab	0.81 ± 0.96 a	10.98 ± 0.18 a	339.51 ± 0.06 a	50.30 ± 0.13 a
	N1	17.11 ± 1.90 c	0.40 ± 0.02 c	7.56 ± 0.44 c	315.94 ± 3.36 bc	44.93 ± 0.50 bc
	N2	14.28 ± 0.48 d	0.30 ± 0.02 d	6.02 ± 0.22 d	290.69 ± 6.16 d	42.90 ± 0.67 c
	SN0	22.03 ± 0.34 a	0.67 ± 0.05 b	9.76 ± 0.38 b	324.26 ± 2.86 b	49.27 ± 0.23 a
	SN1	20.98 ± 0.70 ab	0.62 ± 0.05 b	9.66 ± 0.42 b	336.94 ± 0.25 a	49.83 ± 3.07 a
	SN2	18.91 ± 0.22 bc	0.43 ± 0.04 c	7.80 ± 0.48 c	310.65 ± 1.80 c	48.20 ± 0.75 ab

Note: CK0 (control treatment), N1 (50 mmol L^−1^ NaCl), N2 (100 mmol L^−1^ NaCl), SN0 (water + S-ABA spray at the stage of young panicle differentiation), SN1 (50 mmol L^−1^ NaCl + S-ABA spray at the stage of young panicle differentiation), SN2 (100 mmol L^−1^ NaCl + S-ABA). Different lowercase letters indicate significant differences between different treatments of the same rice variety (*p ≤* 0.05).

## Data Availability

The datasets used and/or analyzed during the current study are available from the corresponding author on reasonable request.

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
