# Peer review of "S-ABA Enhances Rice Salt Tolerance by Regulating Na+/K+ Balance and Hormone Homeostasis"

_metabolites, 2024, doi:10.3390/metabo14040181_

Round 1
Reviewer 1 Report
Comments and Suggestions for Authors
The manuscript by Jiang et al. is devoted to an important problem: the effect of S-ABA on rice growth and development under salt stress. This work contains very interesting results, but I have some comments to the manuscript:
1. The Abstract section is very difficult to read. Some sentences are incomplete. The second sentence is too long; It's very difficult to follow the meaning. The Abstract needs to be improved. Authors should also carefully read and revise the entire text of the manuscript.
2. M&M section:
P. 4, lines 149-174: These paragraphs contain only information about other articles and do not describe the methods used. The MM section should contain at least a brief description of each method.
3. The Results section (Figures):
Some error bars are not visible on the figures. Some figures have text that is difficult to see. In Figure 7, the description of the meaning of * and ** (at the bottom of the figure) is not completely visible.
On the whole, the authors should be more attentive to the presentation of the information, in particular, to formulate sentences more accurately and make logical connections. The quality of all figures should be improved. The manuscript needs significant revision.
Reviewer 2 Report
Comments and Suggestions for Authors
While this paper demonstrates strong potential, there may be room for enhancement in certain areas to align it more closely with the publication criteria of this journal. Otherwise, I have a lot of recommendations to increase the quality of your manuscript. Be careful with the writing and mistakes.
I think that this is one of the articles with so many grammatical mistakes that I have never corrected, please, fix all of them.
Otherwise, this text discusses an effect of abscisic acid (ABA) study on rice under salt stress. Salt stress has been shown to inhibit rice growth, affect the photosynthesis of rice leaves, and disturb the ion balance of plants.
In order to make the article more interesting, a more in-depth discussion on the consequences could be included.
I have found a lot of grammatical mistakes, but I am not going to tell you all of them. You must read again your paper one more time very slowly and correct all of them. Some of them are which follows in the next pages.
I really think that you have avoid the use of the template of this journal because I can not find the logo of MDPI at the top of this paper. Please, download the template and use it.
Line 11. You have started the sentence with the word “It” in bold. In this journal you must avoid the use of bold in the main text.
Line 12. I have found that in the text there are a lot of grammatical errors. One of them, and I have found many more that you must look for them, is the next one. Just after the word “Acid” you must write a space. And just after the first bracket you must avoid the space. This is simple grammatical mistake but very common in your whole paper.
Line 14. Just after the number you must write a space. Just follow the rules of this journal. You must download a published article of this journal and copy its style, is very easy if you do not want to read the rules of this journal.
Line 18. I do not understand why you write some words in bold. You must avoid the use of bold words.
Line 19. You must explain in brackets all the acronyms at least the very first time that you use it in the text. This is a very common mistake in your manuscript. Please, fix it. So, you must read thoroughly your paper looking for this mistake. You must think about the future readers and write a text as much comprehensible as possible. Please, explain the meaning of “CK0”.
Line 21. You must explain in brackets all the acronyms at least the very first time that you use it in the text. This is a very common mistake in your manuscript. Please, fix it. So, you must read thoroughly your paper looking for this mistake. You must think about the future readers and write a text as much comprehensible as possible. Please, explain the meaning of “MDA”.
Line 27. You must explain in brackets all the acronyms at least the very first time that you use it in the text. This is a very common mistake in your manuscript. Please, fix it. So, you must read thoroughly your paper looking for this mistake. You must think about the future readers and write a text as much comprehensible as possible. Please, explain the meaning of “GA”.
Line 27. You must explain in brackets all the acronyms at least the very first time that you use it in the text. This is a very common mistake in your manuscript. Please, fix it. So, you must read thoroughly your paper looking for this mistake. You must think about the future readers and write a text as much comprehensible as possible. Please, explain the meaning of “IAA”.
Line 32. You must write the keywords in alphabetical order.
Line 32. Just after the word “Keywords” you must write two points.
Line 32. You must write the word “Keywords” in its right size, so you must download the template of this journal and use it to avoid this mistake.
Line 38. The reference must be just before the point, not after it.
Line 39. You must write a space just before the reference. This is a very common mistake throughout your whole paper. Please, look for this mistake and fix it everywhere.
Line 43. Another grammatical mistake is that you have to finish a sentence with a point, not with a comma. Please, fix it.
Line 44. You must write the scientific name in italics because this is a scientific journal.
Line 46. You must write a space just before the reference. This is a very common mistake throughout your whole paper. Please, look for this mistake and fix it everywhere.
Line 46. You must avoid the space between the references. So, you must write “[4,5]” instead of “[4, 5]”.
Line 51. You must write a space just before the reference. This is a very common mistake throughout your whole paper. Please, look for this mistake and fix it everywhere.
Line 51. Always just after the point you must write a space. This is a very common mistake throughout your whole manuscript. Look for this mistake and fix it. You have to read thoroughly your manuscript and look for this kind of grammatical mistakes and fix all of them.
Line 59. You must write a space just before the reference. This is a very common mistake throughout your whole paper. Please, look for this mistake and fix it everywhere.
Line 46. You must avoid the space between the references. So, you must write “[9,10]” instead of “[9, 10]”. So, you have to download a published paper of this journal and copy its style, is very easy.
Line 61. You must write a space just before the reference. This is a very common mistake throughout your whole paper. Please, look for this mistake and fix it everywhere.
Line 66. You must write a space just before the reference. This is a very common mistake throughout your whole paper. Please, look for this mistake and fix it everywhere.
Line 72. You must write a space just before the reference. This is a very common mistake throughout your whole paper. Please, look for this mistake and fix it everywhere.
Line 77. You must write a space just before the reference. This is a very common mistake throughout your whole paper. Please, look for this mistake and fix it everywhere.
Line 82. You must write a space just before the reference. This is a very common mistake throughout your whole paper. Please, look for this mistake and fix it everywhere.
Line 92. I do not understand this big space between the paragraphs. Please, download a published paper of this journal and copy its style. Is very easy.
Line 103. Just after the bracket you have to delete the space.
Line 103. You must write “27.5 cm” instead of “27.5cm”. This is a very common mistake in your whole paper. Please, fix it.
Line 109. Just before the bracket you must write a space. This is a very common mistake in your whole manuscript. Please, look for this mistake throughout your whole paper and fix it, you must read the paper thoroughly several times to fix all these mistakes.
Line 110. Just before the initial bracket you must write a space. This is a very common mistake in your whole manuscript. Please, look for this mistake throughout your whole paper and fix it, you must read the paper thoroughly several times to fix all these mistakes.
Line 113. Just before the initial bracket you must write a space. This is a very common mistake in your whole manuscript. Please, look for this mistake throughout your whole paper and fix it, you must read the paper thoroughly several times to fix all these mistakes.
Line 122. You must write “0.499 g” instead of “0.499g”. You must separate the units from the number. In this page almost all the units are wrong written. you must follow this rule in your whole paper. Please, look for this mistake throughout your whole paper and fix it, you must read the paper thoroughly several times to fix all these mistakes. Please, download a published paper of this journal and copy its style if you do not want to read the rules of this journal. Is very easy.
Line 134. I do not understand this big space between the paragraphs. Please, download a published paper of this journal and copy its style. Is very easy.
Line 203. You must write a space between the paragraph and the Table.
Line 204. You must avoid the space just before the word “Table”. Just follow the rules of this journal. You must download a published paper of this journal and copy its style if you do not want to read the rules of this journal. Is very easy. You must download the template of this journal as well and follow its style.
And I am going to stop telling you this kind of mistakes in your manuscript, but you must read several times your paper and fix all this kind of grammatical mistakes.
Line 343. It is not possible to read the tiny letters of the Figures, you must use more space. Put one before another to read the letters better.
Line 488. The comma at the very beginning of this sentence is a grammatical mistake. This comma must finish the sentence of the line 487 just after the word “stability”.
Line 491. I do not understand this big space between the paragraphs. Please, download a published paper of this journal and copy its style. Is very easy.
Line 500. I do not understand this big space between the paragraphs. Please, download a published paper of this journal and copy its style. Is very easy.
Line 504. I do not understand this big space between the paragraphs. Please, download a published paper of this journal and copy its style. Is very easy.
And finally, all the references do not follow the style of this journal. Just follow the rules of this journal. You must download a published paper of this journal and copy its style if you do not want to read the rules of this journal. Is very easy. You must download the template of this journal as well and follow its style.
You must avoid the bold in the article title.
You must write the year of the publication in bold text.
The volume must be in italics.
You must use the long hyphen between the initial page and the last page of a publication.
Just between the authors you must use “;”.
You must write the doi at the very end of every paper.
You must use the abbreviated for of all the journals.
Otherwise, the authors adequately developed the Introduction, presenting the problems but you must write explicitly the objectives of this paper.
The methods are adequate.
The Discussion is well developed and the data presented are correctly compared with other papers.
The authors are to be congratulated for the results obtained in this article.
Comments on the Quality of English Language
The English is good.
Reviewer 3 Report
Comments and Suggestions for Authors
In this study, the physiological and biochemical response of rice plants, cultivars Chaoyou 1000 and Yuxiangyouzhan, under the harsh conditions of salt stress is discussed. The interest of this investigation is the regulatory capacity of S-ABA in influencing the growth and development of rice.
Research highlights that salt stress causes damage to rice growth, but strategic application of S-ABA acts to counteract its effects. This study provides valuable insights into potential agricultural strategies to improve crop performance under challenging environmental conditions.
The idea of the study is salutary, although I think the number of varieties used is not representative. what were the criteria by which these two varieties were chosen?
It is absolutely necessary to improve (with details about the methods used) the material and method section. The equipment used for the determinations must also be specified.
You should consider creating a legend for the graphics used in the manuscript stating the specificațions.
You entered a conclusion in the talk section (Lane 470), I think you should consider moving it to the Conclusions section, or, rewording this statement to be suitable for conclusions.
Comments on the Quality of English Language
An extensive editing of the manuscript is necessary because it contains quite a few editing errors (e.g. sentences starting with lowercase letters, repeating words...)
